# Genetic analysis of *IFNG-AS1* implicates opposite effects to *Leishmania guyanensis*-cutaneous leishmaniasis: rs4913269 confers protection while rs7134599 enhances susceptibility and correlates with high plasma IL-4 and IL-10 levels

Marcus Vinitius de Farias Guerra[1,2], Josué Lacerda de Souza[3,4], Lener Santos da Silva[3,4], José do Espírito Santo Júnior[3,5], Tirza Gabrielle Ramos de Mesquita[1,2], Krys Layane Guimarães Duarte Queiroz[5], George Allan Villarouco da Silva[5], Mauricio Morishi Ogusku[6,7], Mara Lúcia Gomes de Souza[1], José Pereira de Moura Neto[5], Aya Sadahiro[5,7], Rajendranath Ramasawmy [1,2,3,4,5,7]*

1 Fundação de Medicina Tropical Doutor Heitor Vieira Dourado, Manaus, Brazil, 2 Programa de Pós-Graduação em Medicina Tropical, Universidade do Estado do Amazonas, Manaus, Amazonas, Brazil, 3 Faculdade de Medicina, Universidade Nilton Lins, Manaus, Amazonas, Brazil, 4 Programa de Pós-Graduação em Biodiversidade e Biotecnologia da Amazonia Legal (Rede Bionorte), Universidade do Estado do Amazonas, Manaus, Brazil, 5 Programa de Pós-Graduação em Imunologia Básica e Aplicada, Universidade Federal do Amazonas, Manaus, Amazonas, Brazil, 6 Laboratório de Micobacteriologia, Instituto Nacional de Pesquisas da Amazônia, Manaus, Amazonas, Brazil, 7 Genomic Health Surveillance Network: Optimization of Assistance and Research in The State of Amazonas -REGESAM, Manaus, Amazonas, Brazil

* ramasawm@gmail.com

## Abstract

### Background

The long non-coding RNA interferon gamma antisense-1 (*IFNGAS-1)* is essential for Th1 lineage specific expression of *IFNG*. IFN-γ is a key component cytokine in host immune response against intracellular pathogens like *Leishmania*. We investigated the association of two genetic variants of *IFNGAS-1*, rs4913269 and rs7134599, with susceptibility or protection to *Leishmania guyanensis*- induced cutaneous leishmaniasis (*Lg*-CL).

### Methods

A case-control study involving 1,714 individuals (855 *Lg*-CL and 859 healthy controls) was conducted in the state of Amazonas, Brazil. Genotyping of rs4913269 and rs7134599 were performed using direct nucleotide sequencing and polymerase chain reaction-restriction fragment length polymorphism (PCR-RFLP), respectively. Plasma cytokines concentrations (IL-10, IL-12p70, IL-4, IL-1β and TNF-α) were quantified using multiplex Luminex platform. Logistic regression, linkage disequilibrium (LD), and haplotype analyses were applied to assess genetic associations and cytokine correlations.

**Data availability statement:** All the data are in the manuscript and contains all raw data required to replicate the results of the study in the supporting information.

**Funding:** This work was supported by the Brazilian Council for Scientific and Technological Development (CNPq), grant number 404181/2012-0 to RR, Fundação de Amparo e Pesquisa do Estado do Amazonas (FAPEAM), grant numbers 06200151/2020, 01.02.016301.03393/2021-80 and 01.02.016301.01090/2023-94 to RR, and FAPEAM RESOLUÇÃO N. 002/2023-POSGRAD – Coordenador/Auxilio Financeiro. JJ, JS, LS and KLGDQ have fellowships from FAPEAM. JPN and RR are CNPq fellows. The funders had no role in study design, data collection and analysis, decision to publish, or preparation of the manuscript.

## Results

Individuals with the rs4913269 G/G genotype had a 46% reduced risk of developing $Lg$-CL, (OR adjusted for age and sex [$OR_{adj}$] = 0.54; 95% CI 0.39-0.75; $Pv_{adj}$ = 0.0001). Carriers of the rs7134599 A/A genotype had a 130% increased risk of progression to $Lg$- CL ($OR_{adj}$ = 2.3; 95% CI, 1.6–3.4; $P$ = 0.0001). The rs7134599 A/G genotype also showed a 52% increased risk compared to GG genotype ($OR_{adj}$ = 1.52, 95%CI 1.22-1.89; $Pv_{adj}$ = 0.0002). The rs4913269 G/G genotype was associated with lower levels of IL-10 ($P$ = 0.05) and IL-12p70 ($P$ = 0.009) compared to the C/C genotype. Conversely, the rs7134599 AA genotypes were correlated with higher levels of TNF-α, IL-4, IL-10 and IL-1β in comparison to the GG genotype. LD revealed independent segregation of the variants.

## Conclusions

The *IFNG-AS1* variants rs4913269 and rs7134599 exert opposing effects on $Lg$-CL risk and modulate key cytokines involved in disease pathogenesis. These findings underscore the regulatory role in immune responses and increase our understanding of the immunogenetic basis of CL and support the potential *IFNG-AS1* as a biomarker for susceptibility.

### Authors summary

Leishmaniasis is a disease caused by the protozoan parasites *Leishmania* that occurred when the infected sandfly phlebotomine injected the parasite during blood meals. Cutaneous leishmaniasis causes skin lesions. In regions where the parasite is present, many people do not develop the disease. Understanding why some people develop the disease and others don't, can help in designing vaccine or new therapy such as immunotherapy. Furthermore, understanding how the system of defense (immunological defense) of individuals to the parasite *Leishmania* works can also help us to know why some individuals are susceptible to developing the disease, while others are protected to the parasite. In this work, we examined the genetics of the system of immune defense of individuals who develop the disease and those who do not develop the disease. We studied two genetic variations of the gene *IFNG-AS1* and showed that one genetic variant is associated with disease development and one with protection.

## Background

Leishmaniasis is a vector-borne disease caused by the flagellated protozoan *Leishmania spp*. The extracellular promastigote form is transmitted through the bites of phlebotomine sandflies and transforms into the intracellular amastigote form within

the mammalian host following blood meals. Leishmaniasis presents a wide range of clinical manifestations, ranging from asymptomatic cases and self-healing or non-healing cutaneous leishmaniasis (CL) to severe mucosal involvement or life-threatening visceral leishmaniasis (VL). CL is the most common form of human leishmaniasis, characterized by inflammatory ulcerative cutaneous lesions with elevated borders, confined to the skin.

Leishmaniasis is a neglected tropical disease despite it is endemic in 99 countries across Africa, Asia, Southern Europe, the Middle East and South America according to WHO in 2022 [1]. In 2023, 272098 cases of new CL and 11922 new VL cases were reported to WHO [2]. This disease, associated with poverty, malnutrition, population displacement and poor housing conditions, has a significant global burden and affects predominantly vulnerable populations living in remote regions with limited access to healthcare. Leishmaniasis is a parasitic zoonosis, and its transmission patterns are changing with increasing urbanization due to deforestation and human invasion. To date, there is no vaccine and treatment efficacy is low.

The involvement of the immune system in leishmaniasis has been extensively demonstrated in the resistant C57BL/6 and the susceptible BALB/c mouse strains infected with *L. major*. BALB/c mice exhibit a robust T-helper 2 (Th2)-type response to *L. major*, leading to disease progression, in contrast to resistant C57BL/6 mice, which develop a strong T-helper 1 (Th-1)-type response, predominantly dependent on the interleukin-12 (IL-12)/interferon-γ (IFN-γ)/tumor necrosis factor-alpha (TNF-α) axis [3]. IFN-γ plays a critical role in controlling *Leishmania* growth during infection. *IFNG* knock-out C57BL/6 mice are unable to control *L. major* infection. Blocking IFN-γ signaling results in increased lesion size and parasite burden [4].

In Brazil, *L. braziliensis* (*Lb*), *L. guyanensis* (*Lg*), *L. lainsoni*, *L. amazonensis*, *L. shawi*, *L. naiffi* and *L. lindenbergi* are the major species causing tegumentary leishmaniasis. In the state of Amazonas, *Lg* is the primary etiological agent of CL, accounting for approximately 95% of CL cases. In regions where *L. braziliensis* is endemic, individuals with a positive delayed-type hypersensitivity (DTH+) response to *Leishmania* antigen but without a history of disease occurrence have been observed, indicating the presence of subclinical or asymptomatic infection [5].

In regions endemic for leishmaniasis, not all individuals develop disease following bites from *Leishmania*-infected phlebotomine sandflies, suggesting that host genetic factors significantly influence clinical outcomes. Notably, the host-pathogen interaction has several layers of complexity, including the genetic background of the host, the virulence and genotype of the parasite, the *Leishmania*-phlebotomine vector and an environment favorable for the development of lesions. There is evidence of a major recessive gene controlling susceptibility to *Lb*-CL among migrants to an endemic area of leishmaniasis in Bolivia [6]. Family-based genetic epidemiological studies have demonstrated the role of genetic components in controlling susceptibility to *L. peruviana*-CL, particularly at an early age of onset in Peru [7]. Familial aggregation of CL and ML, caused by *L. braziliensis*, in endemic regions supports the hypothesis of heritability and genetic susceptibility of clinical forms of leishmaniasis [8]. Genetic variants within the *IFNG* are suggested to influence cytokine levels of IFN-γ [9–11]. Our findings indicate that the *IFNG* variant rs2069705 may serve as a genetic modifier of the clinical outcome of *L. guyanensis*-infection and individuals with a haplotype of several single nucleotide variants (SNVs) within *IFNG*, associated with low IFN-γ plasma levels, have a 60% increased risk of developing *Lg*-CL [11].

The long non-coding RNA (lncRNA) interferon gamma antisense-1 (*IFNG-AS1)*, also known as *TMEVPG1* and *NEST*, is adjacent to *IFNG* on chromosome 12q14. In a murine model of multiple sclerosis, *IFNG-AS1* was identified as a susceptibility *locus* for Theiler's virus-induced demyelinating disease (TMEV-IDD) [12]. Subsequently, the homologous region on human chromosome 12q14–15 was associated with multiple sclerosis in humans [13]. *IFNG-AS1* is induced in response to Th1 differentiation through mechanisms dependent on Signal Transducer and Activator of Transcription 4 (STAT4) and T-box expressed in T cells (T-bet) [14–16], and *IFNG-AS1* cooperates with T-bet to promote *IFNG* transcription. T-bet binds to the *IFNG-AS1* promoter/proximal enhancer, as well as to upstream distal enhancers in both developing and differentiated effector Th1 cells, playing a critical role in epigenetic remodeling of these regions **[17]**. *IFNG-AS1* is essential for Th1 lineage-specific expression of *IFNG* and is co-expressed with *IFNG*. *IFNG-AS1* contributes

 

to *IFNG* expression, and the human *IFNG-AS1* is selectively expressed in Th1 cells, particularly in effector Th1 cells, and is regulated by the Th1 transcription factors STAT4 and T-bet [17]. *IFNG-AS1* expression is downregulated after *in vitro* stimulation of murine CD4+ or CD8 + splenocytes, while *IFNG* expression is upregulated [17]. T-bet guides the epigenetic remodeling of *IFNG-AS1* proximal and distal enhancers, leading to recruitment of the stimulus-inducible transcription factors nuclear factor-kappa B (NF-κB) and ETS proto-oncogene (Ets-1) to the *locus* [18]. The enhancer-specific activity of *IFNG-AS1* and its transcriptional regulation depend on NF-κB.

Gene expression profiling of colonic tissue surgical samples from patients with ulcerative colitis revealed high expression of *IFNG-AS1* [19], and bioinformatics analysis suggests that the *IFNG-AS1* is associated with the inflammatory bowel disease (IBD) susceptibility *locus* SNV rs7134599 [20]. Moreover, its genomic location is adjacent to the inflammatory cytokine, *IFNG*. The *IFNG-AS1* SNV rs4913269 is associated with susceptibility to *L. braziliensis*-CL [21]. Considering the importance of *IFNG-AS1* in regulating the expression of *IFNG*, this study investigated whether the variants of *IFNG-AS1* rs4913269 C/G and rs7134599 A/G may be associated with *L. guyanensis*-CL in the state of Amazonas, Brazil.

## Materials and methods

### Ethical statement

The study was approved by the Research Ethics Committee of Fundação de Medicina Tropical Heitor Vieira Dourado under approval file number CAAE 09995212.0.0000.0005. This research was conducted in accordance with the Declaration of Helsinki. Written informed consent was obtained from all participants or from parents/guardians for participants under 18 years of age, for the collection of biological samples and subsequent analysis.

### Study population

The present study was conducted in the peri-urban region of Manaus, the capital city of the state of Amazonas, Brazil, where progressive deforestation for settlements, agriculture and farming has been observed over the years, resulting in the emergence of endemic areas for *L. guyanensis* due to human encroachment. This study population has been previously described in multiple studies [22–26]. The research was conducted at the Fundação de Medicina Tropical Dr. Heitor Vieira Dourado (FMT-HVD), Manaus, Amazonas, Brazil. Treatment-naïve patients with active confirmed CL caused by *L. guyanensis* for the first time and presenting with six or fewer lesions were included in the study, with most having a single lesion. All patients presented recent lesions, ranging from three to five weeks. Patients with active CL were recruited between November 2012 and April 2017 at FMT-HVD, a regional reference center for tropical diseases. Healthy controls (HCs) were recruited from the same endemic area as the patients and underwent a thorough clinical examination by physicians to exclude any prior history of leishmaniasis. They shared similar socioeconomic and epidemiological backgrounds as the patients with active CL and had been living in the endemic area for more than five years. Most participants in this study were agricultural or farm workers. A significant majority of the study participants had a history of *Plasmodium vivax* malaria, as these regions also are endemic for malaria. All participants tested negative for HIV infection and had no history of cardiovascular, renal, or diabetic disease. Pregnant women were excluded. HCs were not stratified as asymptomatic or non-infected groups, as no delayed hypersensitivity test (Montenegro skin test) was performed for *Leishmania* antigens. This population is an admixed group of Native American ancestry, commonly referred to as "caboclo", comprising 50–60% Native American, 40–50% European and approximately 10% African ancestry [27].

### Sample size estimation

The effective sample size for genetic association analysis for case-control study, assuming multiple gene inputs for a trait, was calculated based on several assumptions, including a minor allele frequency of 5%, disease prevalence of 5%, complete linkage disequilibrium (LD) between the marker and the trait, a case–control ratio of 1:1, a type 1 error rate of 5%

and an odds ratio of 1.5 and 2.0 for heterozygotes and homozygotes, respectively, with statistical power of 80%, using the Genetic Power calculator of Harvard University. The genetic allelic model indicated a required sample size of 789 cases and 789 controls. This study consisted of 1714 unrelated individuals (855 patients with Lg-CL and 859 HCs). This case–control study compared unrelated patients with *Lg*-CL to healthy unrelated individuals and adhered to the guidelines for Strengthening the Reporting of Genetic Association studies (STREGA).

## Biological materials

### Identification of the *Leishmania* spp

All patients with CL provided biopsy specimens from cutaneous ulcer lesions for the identification of *Leishmania* spp. Parasite presence was confirmed by microscopic-examination of Giemsa-stained lesion scarifications. DNA extracted from the biopsy specimens was subjected to *Leishmania vianna*-specific PCR to distinguish *L. guyanensis* from *L. braziliensis,* following previously established protocols [28,29]. Species identification was confirmed via direct nucleotide sequencing using an ABI 3130XL DNA genetic analyzer. A 233-bp fragment of the heat shock protein 70 (HSP70) gene and a 227-bp fragment of the mini exon genes were sequenced as described previously [26]. Notably, all *Lg*-CL patients included in this study were also enrolled in our previous studies [22–26].

### DNA purification and genotyping of IFNGAS1 variants

All participants provided 5 mL of peripheral blood via venipuncture, collected in ethylenediaminetetraacetic acid (EDTA) coated Vacutainer tubes containing (Becton Dickinson, São Paulo, Brazil). Genomic DNA was extracted using the proteinase K digestion and salting-out method [30].

### Genotyping of IFNGAS1 variants

The rs4913269 and rs7134599 variants of the *IFNGAS-1* were genotyped using direct nucleotide sequencing and polymerase chain reaction-restriction fragment length polymorphism (PCR-RFLP), respectively. The following primers pairs were used: rs4913269F: 5'-TTT TAC CCC TCG CTC CCC T-3' and rs4913269R: 5'-CCA ACC CAA ATG CCC ATC CA-3' for (rs4913269) and rs7134599F: 5'-CCC TTT CCA TTT CTA CTC TAG GC-3' and rs7134599R: 5'-ATG AGC TGG CTT CTA AGG AAT GG-3' for (rs7134599) to amplify the regions encompassing the variants separately.

 PCR was conducted in a 25µL reaction mixture containing 2.0 mmol/L MgCl2, 0.2 pmol/L each forward and reverse primer, 40 µmol/L of each dNTP, 50 ng of genomic DNA and 1 U of Taq polymerase (Invitrogen) in a buffer containing 500 mmol/L KCl and 100 mol/L Tris-HCL (pH 8.3). The PCR cycling conditions for rs4913269 were as follows: an initial denaturation step of 5 min at 95ºC, followed by 35 cycles of denaturation at 95ºC for 15 s, annealing at 62ºC for 15 s and extension at 72ºC for 30 s, with a final extension step at 72ºC for 7 min. The PCR products sizes were 285 bp for rs4913269 and 191 bp for rs7134599.

### Genotyping of rs7134599 by PCR-RFLP.

A 10-uL aliquot of the PCR product was digested with the restriction enzymes *Nla* III or *Hpy*CH4IV (New England Biolabs). *Nla* III cleaves the 19-bp fragment into 123 and 68-bp fragments in the presence of the A allele and the G allele remains uncut. *Hpy*CH4IV cleaves the fragment into 119 and 72 bp fragments in the presence of the G allele, while the A allele remains uncleaved. The PCR-RFLP products were size-separated by 3% agarose gel electrophoresis.

### Genotyping of rs7134599 by direct nucleotide sequencing

The PCR products were purified using a 20% polyethylene glycol (PEG) solution to remove residual primers and free nucleotides. Nucleotide sequencing was performed using the BigDye Terminator v3.1 Cycle Sequencing Kit (Applied

Biosystems) with the same forward or reverse primers used in the PCR reaction. The sequencing was carried out on an ABI 3130XL DNA sequencer using POP-7 polymer. Sequences were initially analyzed using the Sequencing Analysis software (Applied Biosystems, v5.3.1) and only high-quality sequences data were used for SNV analysis.

### Measurement plasma cytokines

Plasma samples were collected from 400 patients with *Lg*-CL and 400 HCs for cytokine analysis [31]. Concentrations of TNF-α, IFN-γ, IL-4, IL-10, IL-6 and IL-12p70 in 5uL of plasma were measured using the Human Cytokine Grp I Panel 27-Plex kit (Bio-Rad, USA) through a multiplex bead-based assay. The assay was conducted following the manufacturer's instruction on the Bio-plex 200 Protein Array System (Luminex Corporation, USA).

### Statistical analysis

Genotypic and allelic frequencies were determined by direct counting. Hardy–Weinberg equilibrium was assessed by comparing observed and expected genotypes distribution using the $\chi^2$ test. Statistical comparisons of genotype distributions between individuals with *Lg*-CL and HCs were performed using R software (version 4.3.1) with the SNPassoc package (version 2.1-0), employing logistic regression. The effects of genotypic variation on circulating plasma cytokine levels were assessed using the Generalized Linear Model (GLM) for quantitative traits in R software with the SNPassoc package. Cytokine data were visualized using the ggplot2 package. Post-hoc analysis following ANOVA was conducted using the rstatix package (version 0.7.2) in R software (r-project.org) for multiple comparisons. Post-hoc P-values were adjusted for multiple testing using the False discovery rate (FDR) correction. Linkage disequilibrium (LD) analysis and LD visualization were performed using Haploview software (version 4.2).

## Results

The study population consisted of 1,714 unrelated individuals, including 855 patients with Lg-CL and 859 HCs. A total of 639 were male patients with *Lg*-CL (mean age ± standard error of the mean 34.29 ± 0.53 years) and 216 were female patients with *Lg*-CL (37.19 ± 1.05 years). In the HCs group, 591 were male (42 ± 0.72 years) and 268 were female (40 ± 1.04 years). Male HCs were significantly older than male patients with *Lg*-CL (p < 0.0001).

The *IFNG-AS1* gene, located on chromosome 12q14, is adjacent to *IFNG*. *IFNG-AS-1* is approximately 165 kb upstream of the *IFNG* coding region on the opposite strand to *IFNG*, spanning nucleotide 67,989,447–68,234,686 covering *IFNG, IL26* and part of *IL22*, as shown in Fig 1. The variant rs4913269 is in the third intron at nucleotide 68,014,065 and rs7134599 is positioned at nucleotide 68,106,295 and 48,474 bp upstream of *IFNG*.

### *IFNGAS-1* variants are associated with Lg-CL

Genotypic and allelic frequencies for rs4913269 and rs7134599 are presented in Tables 1 and 2, respectively. No deviation from Hardy-Weinberg equilibrium was observed for either variant in patients with *Lg*-CL or HCs. The rs4913269 G/G genotype was less frequent in patients with *Lg*-CL (9%) compared with HCs (15%). Individuals with the rs4913269 G/G genotypes exhibited a 46% reduced risk of developing *Lg*-CL compared to individuals with rs4913269 C/C genotypes (odds ratio adjusted for age and sex [$OR_{adj}$]= 0.54; 95% confidence interval [CI], 0.39–0.75; $Pv_{adj}$ = 0.0001). The Akaike Information Criterion (AIC) indicated that the recessive genetic model provided the best fit model in logistic regression analyses.

The distribution of *IFNG-AS1* rs7134599 genotypes differed significantly between patients with *Lg*-CL and HC (P = 2.9 x10$^{-6}$) as shown in Table 2. There was an excess of genotypes A/A and A/G among the patients with *Lg*-CL. Individuals with the A/A genotype exhibited a 130% increased risk of progressing to *Lg*-CL compared to individuals with the C/C genotype ($OR_{adj}$ = 2.3; 95% CI, 1.6–3.4; P = 0.0001), whereas heterozygous A/G individuals had a 50% increased risked (ORadj = 1.5; 95% CI, 1.2–1.9; P = 0.0002). The AIC analysis identified the dominant genetic model as the best-fit model, suggesting that the A allele is associated with increased risk of *Lg*-CL (OR=1.55 [1.32 – 1.82]; P < 0.0001).

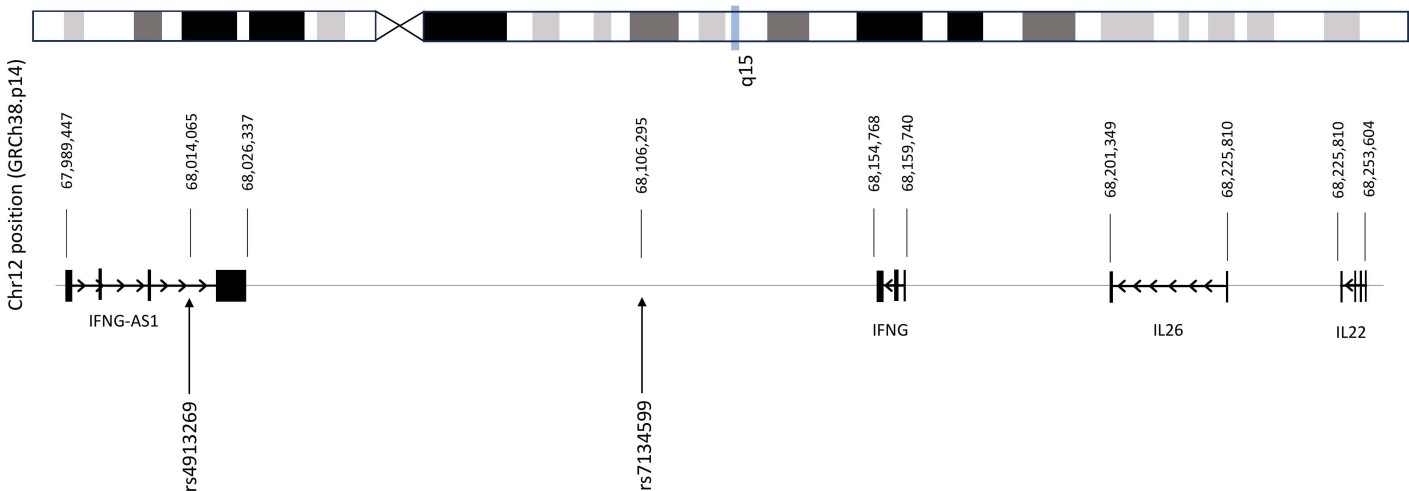

**Fig 1. Genomic position of *IFNGAS-1* and cluster of cytokines genes *IFNG*, *IL26* and *IL22* on human chromosome 12q14-15.** Location of the variants rs4913269 and rs7134599 of *IFNGAS-1* on chromosome 12.

**Table 1. Statistical comparison of genotypic and allelic frequencies of the variant rs4913269 of *IFNG-AS1* using R software version 4.3.1 with the SNPassoc package comparing patients with *Leishmania guyanensis*-cutaneous leishmaniasis and healthy controls from the same endemic areas.**

| Genetic Model | Cases N=774(%) | HC N=806(%) | OR [95% IC] | ORadj [95% IC] | *P*adj value | AIC |
|---|---|---|---|---|---|---|
| Codominant | | | | | | |
| C/C | 365 (47) | 329 (41) | 1 | 1 | 0.001 | 2120 |
| C/G | 337 (44) | 352 (44) | 0.86 [0.70-1.07] | 0.88 [0.71 - 1.1] | 0.17 | |
| G/G | 72 (09) | 125 (15) | 0.52 [0.37-0.72] | 0.54 [0.39 - 0.75] | 0.0001 | |
| Alleles | | | | | | |
| C | 1067 (69) | 1010 (63) | | | | |
| G | 481 (31) | 602 (37) | | | | |
| Dominant | | | | | | |
| C/C | 365 (47) | 329 (41) | 1 | 1 | 0.025 | 2126 |
| C/G+G/G | 409 (53) | 477 (59) | 0.77 [0.63-0.94] | 0.80 [0.65 - 0.97] | | |
| Recessive | | | | | | |
| C/C+G/C | 702 (91) | 681 (85) | 1 | 1 | | |
| G/G | 72 (09) | 125 (15) | 0.56[0.41-0.76] | 0.57 [0.42 – 0.78] | 0.0004 | 2119 |
| Overdominant | | | | | | |
| C/C+G/G | 437 (56) | 454 (56) | 1 | 1 | | |
| G/C | 337 (44) | 352 (44) | 0.99 [0.82-1.21] | 1.0 [0.82 – 1.24] | 0.92 | 2131 |

Abbreviations: Cases, patients with cutaneous leishmaniasis; HCs, Healthy controls; CI, confidence interval; OR, odds ratio; ORadj, odds ratio adjusted for age and sex; *P*adj, *P*-value adjusted for age and sex

**Table 2. Statistical comparison of genotypic and allelic frequencies of the rs7134599 variant of *IFNG-AS1* between patients with *Leishmania guyanensis*-cutaneous leishmaniasis and healthy controls from the same endemic areas.**

| Genetic Model | Cases N = 747 (%) | HCs N = 804 (%) | OR [95% IC] | OR$_{adj}$ [95% IC] | *P*adj value | AIC |
|---|---|---|---|---|---|---|
| Codominant | | | | | | |
| G/G | 366 (49) | 490 (61) | 1 | 1 | 2.94e-06 | 2069 |
| A/G | 301 (40) | 268 (33) | 1.44 [1.17-1.79] | 1.52 [1.22 – 1.89] | 0.0002 | |
| A/A | 80 (11) | 46 (6) | 2.33 [1.58-3.43] | 2.25 [1.52 – 3.33] | <0.0001 | |
| Alleles | | | | | | |
| A | 461 (31) | 360 (22) | | 1.55 [1.32 – 1.82] | <0.0001 | |
| G | 1033 (69) | 1248 (78) | | | | |
| Dominant | | | | | | |
| G/G | 366 (49) | 490 (61) | 1 | 1 | | |
| A/G+A/A | 381 (51) | 314 (39) | 1.57 [1.28-1.93] | 1.63 [1.33 – 2.00] | 2.98e-06 | 2070 |
| Recessive | | | | | | |
| G/G+A/G | 667 (89) | 758 (94) | 1 | 1 | | |
| A/A | 80 (11) | 46 (6) | 2.01[1.38-2.94] | 1.90 [1.29 – 2.79] | 8.48e-04 | 2081 |
| Overdominant | | | | | | |
| G/G+A/A | 446 (60) | 536 (67) | 1 | 1 | | |
| A/G | 301 (40) | 268 (33) | 1.30 [1.05-1.60] | 1.37 [1.11 – 1.69] | 3.85e-03 | 2084 |

Abbreviations: Cases, patients with cutaneous leishmaniasis; HCs, Healthy controls; CI, confidence interval; OR, odds ratio; ORadj, odds ratio adjusted for age and sex; *P*adj, *P*-value adjusted for age and sex

We have previously demonstrated that the T allele of the rs2069705 C/T variant of the *INFG,* located in the promoter region at -1616 bp, is significantly associated with susceptibility to *Lg*-CL in the same study population [11]. This variant is located at nucleotide position 68,161,231 on the chromosome 12, approximately 55 Kb downstream of the *IFNG-AS1* rs7134599 variant. Linkage disequilibrium (LD) analysis among the three variants was conducted using the Haploview 4.2 program. The $r^2$ and D' values for LD were very low, as shown in Fig 2, indicating that these variants segregate independently.

Haplotype analysis identified eight haplotypes as presented in Table 3. The CAT haplotype (*P*-value = 9.0 X 10$^{-6}$) is associated with susceptibility to *Lg*-CL, while the GGC haplotype (*P*-value = 2.0 X 10$^{-4}$) is associated with protection.

### Plasma cytokines levels by genotypes of *IFNG-AS1* rs4913269 and rs7134599

For cytokine analysis, 354 patients with Lg-CL (264 males and 90 females) and 376 healthy controls (269 males and 107 females) were included in the study and shared the same socio-epidemiological and ecological characteristics. The average age of the male patients and healthy controls were 39.8 ± 1.57 and 45.2 ± 1.58 years old, respectively. Th average age of the female patients and healthy controls were 34.8 ± 0.80 and 43.7 ± 1.8 years old, respectively. All patients presented recent lesions, ranging from three to five weeks, and were treatment naïve.

Expression of *IFNG-AS1* has been shown to positively correlate with IL-6 levels, and negatively with IL-10 levels [32]. Additionally, elevated expression of *IFNG-AS1* has been associated with activation of IFN-γ, IL-1, IL-6, and TNF-α in ulcerative colitis specimens [17]. The rs7134599 variant has also been implicated with asthma [33]. IL-4 plays a pivotal role in asthma pathogenesis. Plasma levels of TNF-α, INF-γ, IL-4, IL-10, IL-6 and IL-12p70 stratified by rs4913269 and rs7134599 genotypes among patients with *Lg*-CL and HCs are shown in S1 Fig. Notably, we previously demonstrated that plasma levels of TNF-α, INF-γ, IL-4, IL-10, IL-6 and IL-12p70 were significantly elevated in patients with *Lg*-CL compared to HCs [31]. Fig 3 presents cytokines levels by genotypes of rs4913269 across all subjects (patients with *Lg*-CL and

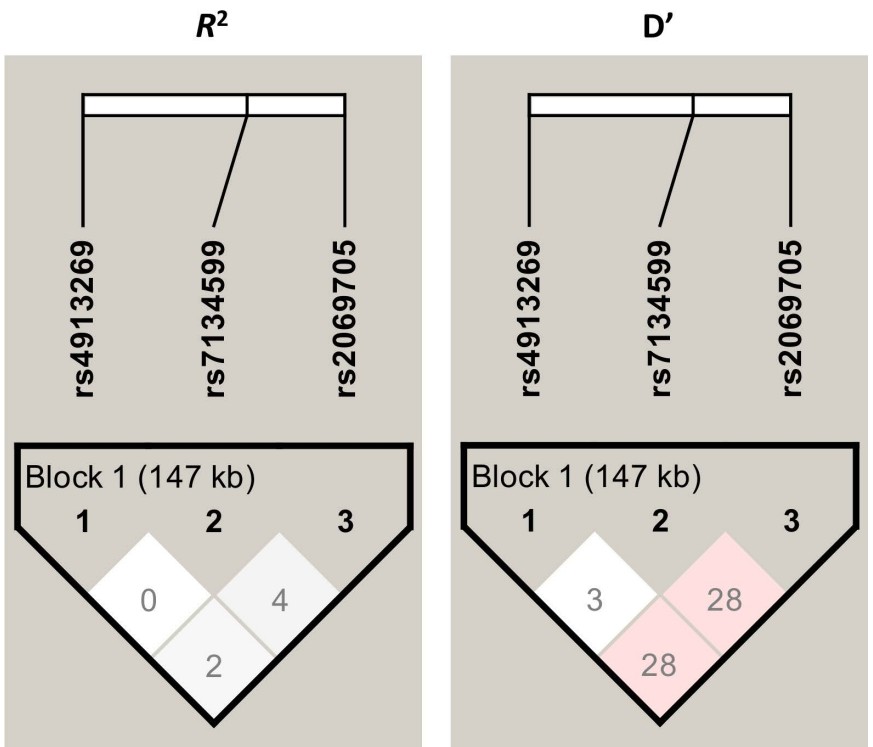

**Fig 2. Linkage Disequilibrium (LD) among rs4913269 (*IFNGAS1*), rs7134599 (*IFNGAS1*), and rs2069705 (*IFNG*) variants.** The LD plot was performed using Haploview 4.2 and displays $R^2$ and D' measures.

**Table 3. Distribution of the *IFNG-AS1* Haplotypes including the rs2069705C>T of *IFNG* in the Study Population.**

| Haps | SNV 1 | SNV 2 | SNV 3 | Freq. total % | Cases n 1,148 (%) | HC n 1,202 (%) | $X^2$ | OR | CI 95% | p.Value |
|---|---|---|---|---|---|---|---|---|---|---|
| 1 | C | G | C | 29 | 27 | 30 | 2,7 | 0.86 | 0.72-1.02 | 0.09 |
| 2 | C | G | T | 20 | 21 | 19 | 2 | 1.16 | 0.95-1.42 | 0.15 |
| 3 | G | G | C | 19 | 16 | 22 | 14 | 0.68 | 0.55-0.83 | 2.0E-4 |
| 4 | C | A | T | 11 | 14 | 8 | 19 | 1.8 | 1.39-2.44 | 8.8E-6 |
| 5 | C | A | C | 6,5 | 7 | 5 | 3.8 | 1.4 | 0.99-1.94 | 0.05 |
| 6 | G | G | T | 6 | 5 | 6 | 1.7 | 0.80 | 0.56-1.19 | 0.19 |
| 7 | G | A | C | 4,5 | 5 | 4 | 0.70 | 1.18 | 0.79-1.70 | 0.40 |
| 8 | G | A | T | 4 | 4 | 4 | 0.25 | 0.90 | 0.60-1.40 | 0.61 |

Abbreviations: SNV, Single nucleotide variant; 1, rs4913269 C > G; 2, rs7134599 G > A; 3, rs2069705C>T; χ2, Chi-square test; OR, odds ratio; CI, confidence Interval.

HCs). ANOVA analysis revealed a statistically significant difference in IL-12p70 levels among the (*P* adjusted for age and sex = 0.04). Post-hoc analysis demonstrated that significantly lower levels of IL-10 (*P* = 0.05) and IL-12p70(*P* = 0.009) were correlated with the rs4913269 G/G genotype compared to the C/C genotype across all subjects.

Additionally, the ANOVA test identified significant differences in plasma levels of TNF-α (*P*adj = 0.03), IL-4 (*P*adj = 0.03), IL-10 (*P*adj = 0.01) and IL-1β (*P*adj = 0.02) among the rs7134599 genotypes across all subjects (patients with *Lg*-CI and HCs), as shown in Fig 4. Post-hoc analysis revealed that the A/A genotype was associated with higher TNFα (*P* = 0.06

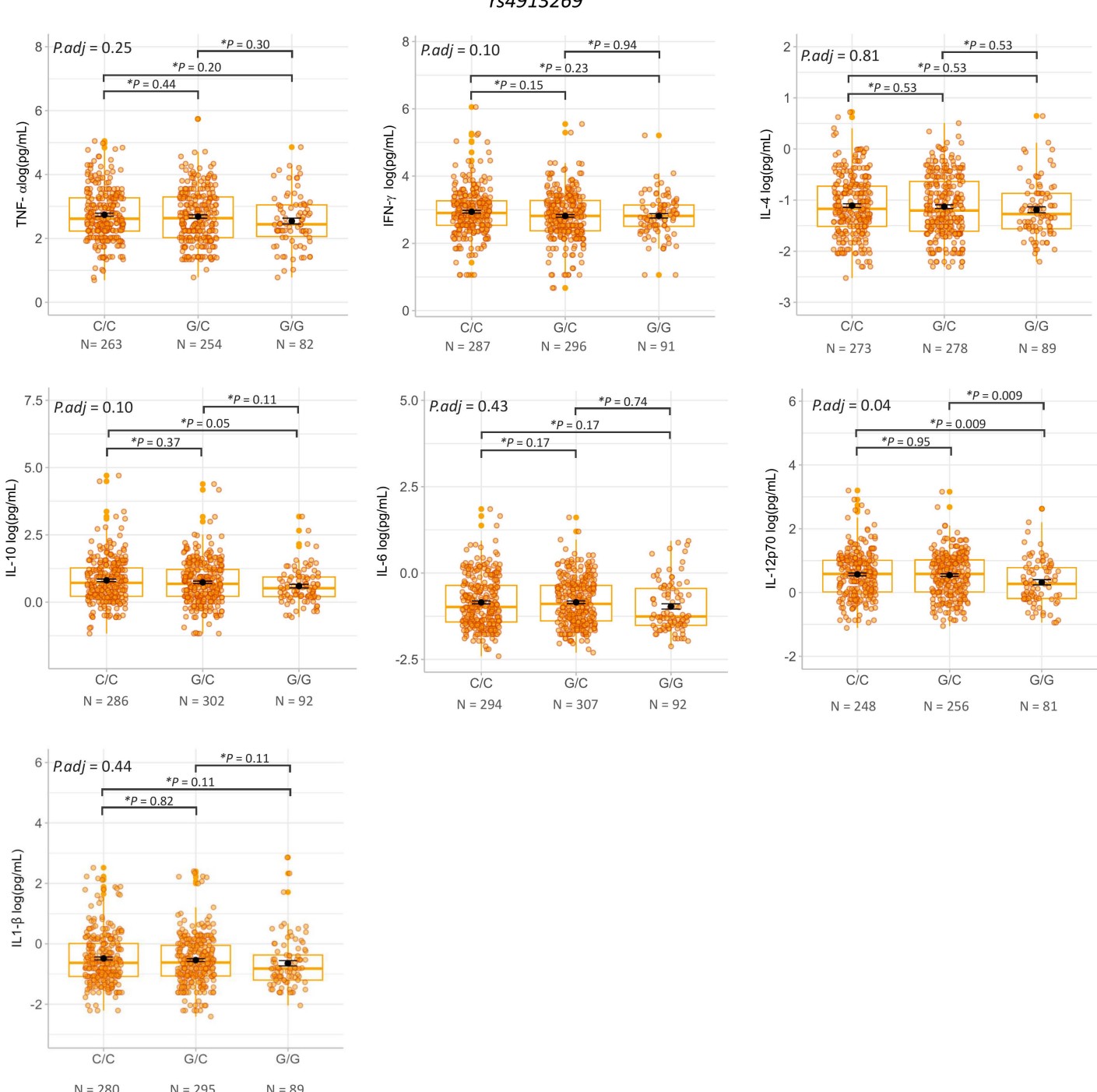

**Fig 3. Analysis of plasma cytokines levels of TNF-α, IFN-γ, IL-4, IL-10, IL-6, IL-12p70 and IL-1β by genotypes of variant rs4913269 across all subjects (patients with *Lg*-CL and HCs).** Statistical analysis was performed using the ANOVA test with *P*-value adjusted for sex and age (*P*adj) for distribution among genotypes and post-hoc test for pairwise comparison between genotypes (*P* = p-corrected for false discovery rate (FDR)).

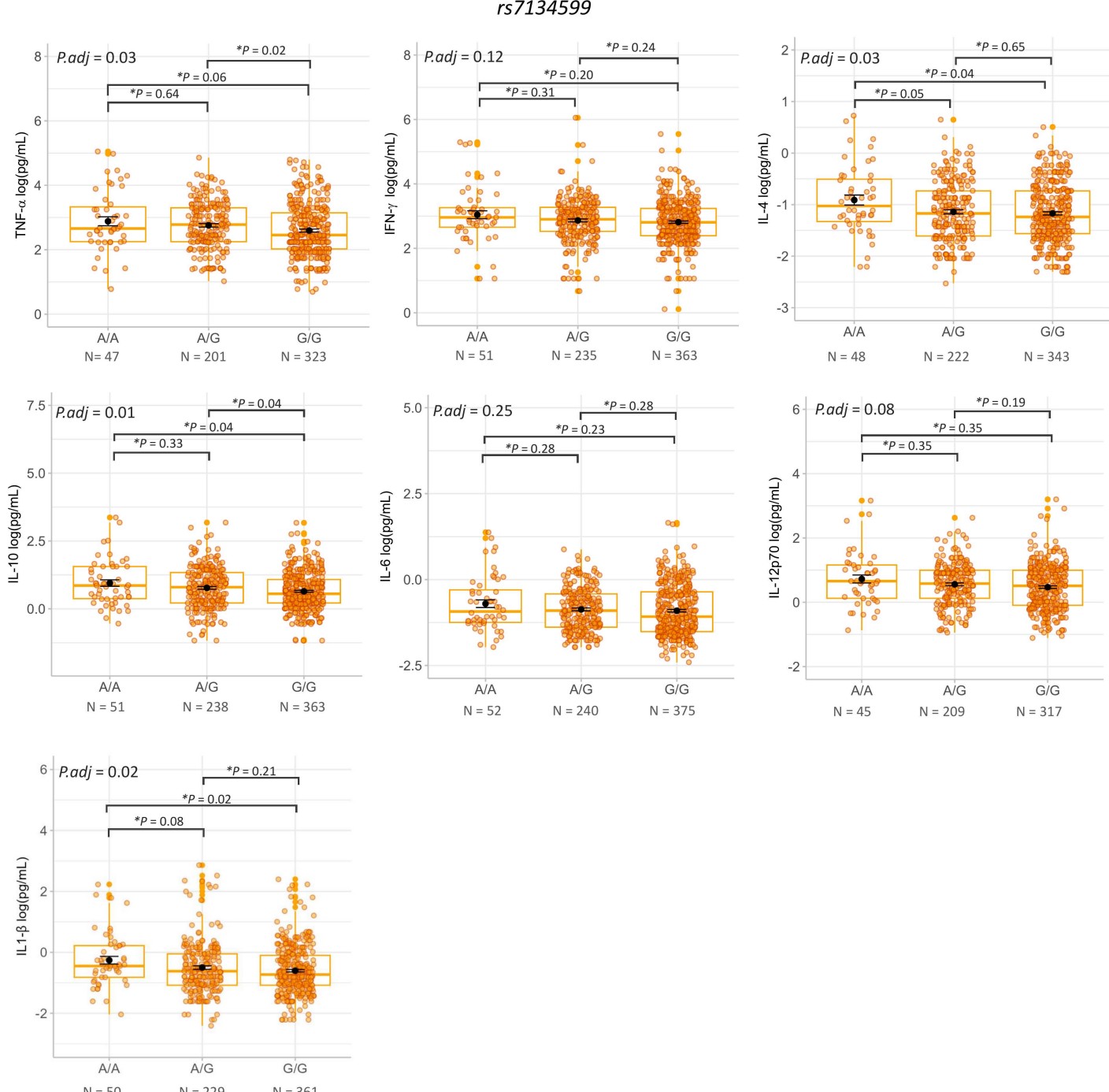

**Fig 4. Analysis of plasma cytokines levels of TNF-α, IFN-γ, IL-4, IL-10, IL-6, IL-12p70 and IL-1β by genotypes of variant rs7134599 across all subjects (patients with *Lg*-CL and HCs).** Statistical analysis was performed using the ANOVA test with *P*-value adjusted for sex and age (*P*adj) for distribution among genotypes and post-hoc test for pairwise comparison between genotypes (*\*P=p*-corrected for false discovery rate (FDR)).

borderline), IL-4 (*P* = 0.04), IL-10 (*P* = 0.04) and IL-1β (*P* = 0.02) compared to the G/G genotype across all subjects, as shown in Fig 4. All *P*-values were corrected by false discovery rates (FDR).

## Discussion

Individuals from the same endemic areas are more likely to share similar socio-epidemiological and ecological characteristics. However, only a subset of these individuals develops *Lg*-CL, highlighting the role of host genetic factors in disease susceptibility. To advance the understanding of the molecular mechanisms underlying the immunopathogenesis of *Lg*-CL, we conducted a case-control genetic analysis of *IFNG-AS1* variants.

Previous studies have indicated that *IFNG-AS1* positively regulates *IFNG* expression in immune cells such as CD8+ and CD4 + T cells [14–16]. IFN-γ is a key immunoregulatory cytokine produced by T cells and innate lymphoid cells, playing a pivotal role in host-defense against invading pathogens. LncRNA, including *IFNG-AS1,* are sequences exceeding 200 nucleotides in length, transcribed from genes lacking open reading frames for protein translation. They are involved in transcriptional regulation by modulating RNA polymerase II activity, promoter activation [34], chromatin remodeling, telomere activity, subcellular structural organization [35], and post-transcriptional mRNA processing, including splicing, editing, transport, translation, and degradation [36].

In this study, we identified distinct roles for two *IFNG-AS1* variants in modulating susceptibility to *Lg*-CL. Notably, individuals homozygous for the G allele at rs4913269 in the LncRNA *IFNG-AS1* appear to be protected from developing *Lg*-CL, in contrast to findings in *Lb*-CL where the same G allele was associated with disease susceptibility [21]. Peripheral blood mononuclear cells from heterozygous (A/G) patients with *Lb*-CL, stimulated with soluble *Leishmania* promastigote antigens, showed a significantly higher percentage of IFN-γ and TNFα–producing T cells compared with G/G homozygotes, suggesting that G/G homozygotes are at higher risk of developing *Lb*-CL [21]. However, no significant differences of IFN-γ and TNFα–producing T cells was observed between the AA genotypes and GG homozygotes [21].

We observed that homozygous individuals for the rs4913269 G allele exhibited lower plasma levels of IL-10 and IL-12p70. IL-12 is essential for the expression of *IFNG* and IFN-γ plays a central role in host-defense against intracellular pathogen. In *IL10*-knockout mice with spontaneous colitis, elevated expression of *IFNG-AS1* was noted [17]. The protective effect of rs4913269 is noteworthy, particularly considering the prior findings in *Lb*-CL where the same allele was linked to susceptibility [21]. The rs4913269 G allele may promote a controlled Th1 response, suppressing IL-10 and limiting IL-12p40 as part of a feedback effect of early parasite clearance. Indeed, these immunological profiles may suggest that bearers of the rs4913269 G/G genotype achieve parasite control without excessive inflammation.

IL-10 is an anti-inflammatory cytokine that can inactivate macrophages, inhibit the elimination and multiplication of intracellular pathogens, and impair Th-1 immune responses, including the production of IFN-γ and TNF-α and avoid tissue damage. IL-12-p70, known to promote Th1 differentiation, is produced primarily by dendritic cells and macrophages in response to infection. In rs4913269 G/G carriers, a reduced IL-10 level may preserve macrophage activation and enhance parasite elimination. Meanwhile, lower IL-12p70 may indicate that the immune system achieves sufficient IFN-γ levels early in infection, limiting the need for continued IL-12 driven stimulation to avoid exacerbating proinflammatory reaction, while keeping in check the parasites. This is consistent with the broader understanding that genes involved in immune responses must be precisely coordinated and tightly regulated to ensure rapid expression of multiple genes to control microbial pathogens while mitigating host tissue damage [37]. The *IFNG-AS1* gene is selectively expressed in Th1 effector cells, and we hypothesized that reduced IL-12 and IL-10 levels may facilitate an adequate IFN-γ response, ensuring sufficient macrophage activation to control intracellular pathogens, thereby preventing tissue damage following parasites clearance or control during early infection.

Interestingly, our findings align observations from biopsy lesion specimens of *Lb*-CL patients that exhibited high expression of pro-inflammatory chemokines and cytokines, primarily associated with the activation and migration of monocytes,

polymorphonuclear cells, and Th1 CD4 + T cells [38], as well as elevated cytolytic activity mediated by CD8 + T cells [39]. Additionally, increased frequencies of NK cells have been observed in the peripheral blood cells of *Lb*-CL patients, with these cells expressing higher IFN-γ, TNF-α, granzyme B, and perforin than CD8+ T cells. Granzyme B expression positively correlates with lesion-size, suggesting NK cells contribute to CL immunopathology [40]. In *Lb*-CL, a positive correlation exists between lesion size and the number of CD4 + T cells expressing IFN-γ and TNF-α [41]. In *Plasmodium falciparum*, children whose cells produced higher IFN-γ following stimulation with *Plasmodium falciparum* antigen in vitro were more likely to experience the clinical manifestations of malaria infection [42]. *IFNG-AS1* expression enhances IFN-γ expression in CD8 + T cells, in response to *Salmonella* infection [16]. Elevated *IFNG-AS1* expression has been shown to augment Th1 responses in Hashimoto's Thyroiditis (HT) patients and may contribute to HT pathogenesis [43]. Conversely, the increased *IFNG-AS1* expression has been associated with decreased Th1 cells and increased Treg cells population in experimental autoimmune Myasthenia gravis models [44].

The rs7134599 in *IFNGAS-1* has been suggested as a susceptibility *locus* for the IBD [20]. In a Tunisian population, the rs713599 A allele was associated with protection against asthma, a condition influenced by IL-4 expression [33]. In our study, we found that the A allele is associated with susceptibility to *Lg*-CL and correlates with higher circulating plasma levels of IL-10, IL-4, IL-1β and TNF-α. IL-10 and IL-4 are well-established inducers of Th2 response which has been linked to disease progression in Balb/c mice, contrasting with the protective Th1 response observed in C57Bl/6 mice [3]. Indeed, *Lg*-CL patients with predominant Th2 cytokines (IL-4 and IL-13) tend to develop lesions earlier than those with a Th1-dominant cytokine profile [45]. In *Lb*-CL patients, higher IFN-γ and IL-10 expressions have been observed in late-stage lesions compared with early-stage lesions [46]. Excess IL-1β has been associated with increased disease severity in patients with CL caused by *L. peruviana* [47]. IL-1β can suppress proinflammatory Th1 response by downregulating IL-12 receptor expression and interfering with IL-6-induced STAT1 phosphorylation [48].

It is important to consider that the observed association may also reflect differences in parasite burden and disease kinetics. It is plausible that rs4913269 G/G individuals mount a rapid and effective immune response, resulting in reduced parasite loads and shorter lesion duration. Conversely, rs713599 A carriers may experience delayed parasite clearance, leading to prolonged or more severe clinical manifestations. Future studies assessing parasite load and lesion healing time in relation to genotype will be valuable to validate these hypotheses.

Moreover, comorbid conditions such as malaria, malnutrition or HIV can profoundly influence immune homeostasis and cytokine profiles, influencing disease progression and treatment response. These conditions may interact synergistically with host genetic factors, including *IFNG-AS1* variants, to increase susceptibility. For instance, HIV can induce immuno-suppression and impair a Th1 response, while malaria might skew cytokine production through chronic immune action. The introduction of these variables in future multivariate models will enhance the accuracy of disease risk prediction, uncover gene-environment interactions and offer better clinical assessments in endemic populations. Of note, individuals with HIV were excluded in this study.

From a translational perspective, genotyping of *IFNG-AS1* variants could enable risk stratification, identifying individuals more likely to develop disease upon exposure. For example, individuals with the rs7134599 A/A genotype may benefit from therapies aimed at modulating IL-10 or IL-4 in combination with conventional treatments.

In this study, we selected two SNVs (rs4913269 and rs7134599) based on their reported associations: rs4913269 with *Lb*-CL in Brazil [19], and rs7134599 with IBD [17,18]. The rs4913269 variant has been identified as a strong expression quantitative trait locus (eQTL) for *IFNG-AS1* and a cis-acting eQTL for *IFNG* [19], while rs7134599 has been character-ized as a strong eQTL in IBD [17,18]. The *IFNG-AS1* gene region, located on chromosome 12q14, encompasses the genes *IFNG, IL26* and *IL22* as shown in Fig 1. Previously, we examined 9 SNVs spanning the *IFNG* gene region and demonstrated a similar LD structure between the patients with *Lg*-CL and HCs [11]. To mitigate the risk of spurious asso-ciations, we incorporated the two *IFNG-AS1* SNVs along with eight *IFNG* SNVs to evaluate the LD structure consistency. The LD structure remains consistent between the patients with *Lg*-CL and HCs, suggesting a shared genetic background

between the two groups, in line with STREGA guidelines (S2 Fig). Collectively, these findings indicate that allele frequency differences between the patients with *Lg*-CL and HCs are unlikely to be attributed to population stratification.

Our study population consists of approximately 50% to 60% of Native American, 40% to 50% European and around 10% African ancestry [27]. The minor allele frequency (MAF) for rs4913269 varies from 20% to 50% across different ethnic groups (https://snpinfo.niehs.nih.gov/snpinfo/snptag.html). Notably, the frequency in Bahia (29%) aligns with that observed in the Admixed American population, whereas in Manaus (37%), it approximates the frequency observed in the Japanese population. Regarding rs7134599, the MAF ranges from 10% to 37%, depending on population ethnicity, with our study population exhibiting a MAF of 22%, like that observed in admixed American populations. In light of differences in MAF across ethnic groups, spurious associations may arise in case-control studies. In our study, this risk is minimized as the study population originates from the same endemic areas and the LD structure in this chromosomal region is similar between the patients and HCs group.

Nevertheless, the associations of rs4913269 G and rs7134599 A allele with CL are insufficient to fully explain protection against and susceptibility to *Lg*-CL, respectively. CL is a complex and multifactorial disease, influenced by multiple genetic and environmental factors. The immune response to infection is determined by the individual adaptive T-helper cell responses and disease progression also depends on the host-pathogen interaction which includes the host's genetic background, parasite virulence, the *Leishmania*-vector phlebotomine characteristics and environmental factors.

In summary, our findings demonstrate that rs4913269 and rs7134599 are independently associated with decreased and increased risk of *Lg*-CL, respectively. These insights contribute to our understanding of the molecular basis of the disease and highlight its immunopathogenic mechanism of the disease, paving the way for the development of immuno-therapeutic strategies.

## Supporting information

**S1 Fig. Analysis of plasma cytokines levels by genotypes of variant rs4913269 and rs7134599 in Cases and Controls.** Statistical analysis was performed using the ANOVA test with P value adjusted for sex and age (Padj) for distribution among genotypes and post hoc test for pairwise comparison between genotypes (*P=p value corrected for false discovery rate (FDR)). (PDF)

**S2 Fig. Linkage Disequilibrium structure of the two *IFNG-AS1* rs4913269 and rs7134599 along with eight single nucleotide variants of *IFNG*.** The LD plot was performed using Haploview 4.2 and displays $R^2$ and D' measures. (PDF)

**S1 Data. Raw data for basic characteristics of the study population for calculations of means, standard deviation and standard error of the mean.** (XLSX)

**S2 Data. Raw data of plasma cytokines and genotypes for correlations.** (XLSX)

## Acknowledgments

The authors thank all the participants in the study.

## Author contributions

**Conceptualization:** Marcus Vinitius de Farias Guerra, Rajendranath Ramasawmy.

**Data curation:** Josué Lacerda de Souza, José do Espírito Santo Júnior, Tirza Gabrielle Ramos de Mesquita, Mauricio Morishi Ogusku, Mara Lúcia Gomes de Souza.

**Formal analysis:** José do Espírito Santo Júnior, Tirza Gabrielle Ramos de Mesquita, George Allan Villarouco da Silva, Mauricio Morishi Ogusku, José Pereira de Moura Neto, Aya Sadahiro.

**Funding acquisition:** Rajendranath Ramasawmy.

**Investigation:** Mara Lúcia Gomes de Souza.

**Methodology:** Josué Lacerda de Souza, Lener Santos da Silva, Tirza Gabrielle Ramos de Mesquita, Krys Layane Guimarães Duarte Queiroz, George Allan Villarouco da Silva.

**Supervision:** Rajendranath Ramasawmy.

**Validation:** Tirza Gabrielle Ramos de Mesquita.

**Writing – original draft:** Marcus Vinitius de Farias Guerra, Josué Lacerda de Souza, José Pereira de Moura Neto, Rajendranath Ramasawmy.

**Writing – review & editing:** Marcus Vinitius de Farias Guerra, Aya Sadahiro, Rajendranath Ramasawmy.

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
