## [Decision Letter · Decision Letter 0]

PNTD-D-25-00468

Genetic analysis of IFNG-AS1 implicates opposite effects to Leishmania guyanensiscutaneous leishmaniasis:  rs4913269 confers protection while rs7134599 enhances susceptibility and correlates with high plasma IL-4 and IL-10 levels

Dear Dr. Ramasawmy,

Thank you for submitting your manuscript to PLOS Neglected Tropical Diseases. After careful consideration, we feel that it has merit but does not fully meet PLOS Neglected Tropical Diseases's publication criteria as it currently stands. Therefore, we invite you to submit a revised version of the manuscript that addresses the points raised during the review process.

Please submit your revised manuscript within 30 days Aug 02 2025 11:59PM. If you will need more time than this to complete your revisions, please reply to this message or contact the journal office at plosntds@plos.org. Please include the following items when submitting your revised manuscript:

* A rebuttal letter that responds to each point raised by the editor and reviewer(s). You should upload this letter as a separate file labeled 'Response to Reviewers '. This file does not need to include responses to any formatting updates and technical items listed in the 'Journal Requirements' section below.

* A marked-up copy of your manuscript that highlights changes made to the original version. You should upload this as a separate file labeled 'Revised Manuscript with Track Changes '.

* An unmarked version of your revised paper without tracked changes. You should upload this as a separate file labeled 'Manuscript '.

We look forward to receiving your revised manuscript.

Kind regards,

Vyacheslav Yurchenko, Ph.D.

Academic Editor

Susan Madison-AntenucciSection EditorPLOS Neglected Tropical Diseases

Shaden Kamhawi

co-Editor-in-Chief

Paul Brindley

co-Editor-in-Chief

**Additional Editor Comments:**

First of all, please accept my sincere apologies for the very long processing of this submission. It was particularly difficult to find competent referees. The review process is now over and I am happy to tell you that all three reviewers considered your work to be important for the community and worth reporting in PLoS Neglected Tropical Diseases. I also share their enthusiasm with a caveat that your manuscript needs a few small but crucial clarifications before it can be accepted for publication. Please address all the comments carefully in your revised text and rebuttal letter. I do not envision another round of reviews if all the comments are addressed and discussed.

**Journal Requirements:**

2) We noticed that you used the phrase 'data not shown' in the manuscript. We do not allow these references, as the PLOS data access policy requires that all data be either published with the manuscript or made available in a publicly accessible database. Please amend the supplementary material to include the referenced data or remove the references.

- TM on page: 8.

5) We have noticed that you have uploaded Supporting Information files, but you have not included a list of legends. Please add a full list of legends for your Supporting Information files after the references list.

6) We note that your Data Availability Statement is currently as follows: "All the data are in the manuscript". Please confirm at this time whether or not your submission contains all raw data required to replicate the results of your study. Authors must share the “minimal data set” for their submission. PLOS defines the minimal data set to consist of the data required to replicate all study findings reported in the article, as well as related metadata and methods (https://journals.plos.org/plosone/s/data-availability#loc-minimal-data-set-definition).

7) Please amend your detailed Financial Disclosure statement. This is published with the article. It must therefore be completed in full sentences and contain the exact wording you wish to be published.

1) State what role the funders took in the study. If the funders had no role in your study, please state: "The funders had no role in study design, data collection and analysis, decision to publish, or preparation of the manuscript.".

If you did not receive any funding for this study, please simply state: u201cThe authors received no specific funding for this work.u201d.

8) Please ensure that the funders and grant numbers match between the Financial Disclosure field and the Funding Information tab in your submission form. Note that the funders must be provided in the same order in both places as well. 

9) Please send a completed 'Competing Interests' statement, including any COIs declared by your co-authors. If you have no competing interests to declare, please state "The authors have declared that no competing interests exist". Otherwise please declare all competing interests beginning with the statement "I have read the journal's policy and the authors of this manuscript have the following competing interests.

**Reviewers' comments:**

Reviewer's Responses to Questions

**Key Review Criteria Required for Acceptance?**

**Methods**

-Are the objectives of the study clearly articulated with a clear testable hypothesis stated?

-Is the study design appropriate to address the stated objectives?

-Is the population clearly described and appropriate for the hypothesis being tested?

-Is the sample size sufficient to ensure adequate power to address the hypothesis being tested?

-Were correct statistical analysis used to support conclusions?

-Are there concerns about ethical or regulatory requirements being met?

Reviewer #1: (No Response)

Reviewer #2: Yes the objectives of the study are clearly stated.

The study design is appropriate for the stated objectives. The male healthy control are significantly older than the male patients. Would this be a confounding factor for the data analysis? How was this difference addressed in the data analysis.

While the population is described and has been used for other studies that are referenced in the paper, more details would be helpful for a better understanding of how the immunological data should be analysed.

Data was provided that these participants were presenting with CL for the first time. Some additional information would be helpful about how long the patients have had symptoms of CL. The immune response for patients presenting weeks to months following infection will be different from patients presenting months to years following initial infection. Is it possible to estimate whether these patients represent early infection or later infections that may be resolving? Should this be considered when analysing the data? It would also be useful to know if patients had any other pre-existing conditions or infections. E.g. HIV status.

Yes the sample size is sufficient to ensure adequate power and a higher than required number of participants were enrolled in the study to ensure statistical power.

The statistical analysis appears appropriate to support the conclusions

No concerns about ethical or regulatory requirements.

Reviewer #3: The main objective is clearly articulated, with a clear hypothesis stated. Is points to investigate whether the genetic variants rs4913269 and rs7134599 of IFNGAS-1 may be associated with susceptibility or protection to L. guyanensis CL. The study design is appropriate to address the stated objectives and the authors gave a complete description of the design. The population is clearly described, and is appropriate for the hypothesis being tested. The studied population is well described, and the authors comment that the same population was reported in other articles of the group (cites 20-24). The studied population covers an important number of well selected patients with Lg-CL and HCs, which is very valuable and not easy to reach for this disease. The sample size is sufficient. The authors presented the calculation for an effective sample size for genetic association analysis for case-control study, and reached those numbers of patients and healthy controls. The statistical analysis is well described, complete and results correct for the study. There are not concerns about ethical requirements. Authors provide the number of the approval by the Research Ethics Committee of FMT-HVD. Besides, the obtained written informed consent from the participants.

**Results**

-Does the analysis presented match the analysis plan?

-Are the results clearly and completely presented?

-Are the figures (Tables, Images) of sufficient quality for clarity?

Reviewer #1: (No Response)

Reviewer #2: Yes the analysis presented matches the analysis plan

Figure 3 legend needs clarification of what A and B represent. Just need to add "A" and "B" to the figure legend.

When biopsies were collected from patients, was the parasite burden estimated in these samples? Could this be correlated to immune responses? Are the elevated levels of cytokines directly correlated to higher amounts of parasites within the lesion? This could be a contributing factor in addition to host genetics.

As this paper focusses on IFNG-AS1 and interferon gamma, can the authors provide a figure that shows IFNG-AS1 expression in terms of IFNg levels? Is there a positive correlation in this data set as seen in the literature?

When blood was collected for cytokine analysis, was flow cytometry perfomed to quantitate the amount of IFNg secreted from CD4 and CD8 T cells? This data would be very useful to correlate with IFNg-AS1 expression.

Reviewer #3: The analysis presented match the analysis plan. The authors described the results obtained with the genetic variants of rs4913269 and rs7134599 of IFNGAS-1 using tables and figures. They also showed their association with plasma cytokine production within Figure 3 and supplementary figure 1. The results are completely presented, with the corresponding significance in each case. Figures are clear. Authors should check minor errors, as small lines missing in Fig. 3B.

**Conclusions**

-Are the conclusions supported by the data presented?

-Are the limitations of analysis clearly described?

-Do the authors discuss how these data can be helpful to advance our understanding of the topic under study?

-Is public health relevance addressed?

Reviewer #1: (No Response)

Reviewer #2: The discussion could be expanded to include other aspects that could contribute to clinical presentation, such as parasite burden, kinetics of disease and underlying clinical conditions. Some insight into how the authors envision using host genetics to improve CL treatment or preventative measures would be useful.

Reviewer #3: The conclusions are supported by the data presented. The authors described the limitations of the analysis, commenting that the associations found between the genetic variants of the studied alleles were insufficient to fully explain protection against and susceptibility to Lg-CL. The authors point to the importance of highlighting immunopathogenic mechanism of the disease, in order to develop immunotherapeutic strategies. Authors did not address the public health relevance. I believe that the manuscript has relevance for public health as cutaneous leishmaniasis represents a serious problem with a wide variability of symptoms and different outcomes. Any advance in the area related to prognosis of clinical outcomes will be useful for the management of the infection.

**Editorial and Data Presentation Modifications?**

Reviewer #1: (No Response)

Reviewer #2: Minor revision

Reviewer #3: The section “Authors summary” needs English improvement.

Minor modifications: the abstract shows statistical significances for the results of rs4913569, but no for results about rs7134599. Figure 3B needs revision (line of significant difference). Within Methods section, paragraph spacing needs to be checked. Statistical significance need to be standardized; p= .005 or p=0.005. Minor grammatical errors. Numbers of cites in bold font or not.

**Summary and General Comments**

Reviewer #1: (No Response)

Reviewer #2: Guerra and colleagues present a really interesting and much-needed approach to understanding the wide range of clinical presentation seen for cutaneous leishmaniasis. This is a very complex, multifaceted area and some additional information would be helpful for more accurate interpretation of the data.

Some minor typographical and grammatical errors throughout.

Reviewer #3: The expertise of the group is well known in the area, with several articles published in the last 6 years in prestigious journals related to single nucleotide variants of different genes in the context of L. guyanenis cutaneous leishmaniasis. These reports are based on the same study population, and, from my point of view, in this way the group gives to the research community broad information which results useful for the study of the disease. Discussion of the association of cytokine concentrations, genetic variants and their explanation regarding the onset immune response found in the disease could be extended. For example in the case of the finding of lower levels of regulatory but also inflammatory cytokines as IL-10 together with IL-12p70. Authors could also comment why they believe they did not find changes regarding IFN-gamma plasma production among the different genetic variants.

PLOS authors have the option to publish the peer review history of their article (what does this mean? ). If published, this will include your full peer review and any attached files.

**Do you want your identity to be public for this peer review?** For information about this choice, including consent withdrawal, please see our Privacy Policy .

Reviewer #1: No

Reviewer #2: **Yes: ** Dr Natalie Prow

Reviewer #3: No

**Figure resubmission:**
---

## [Editor Report · Decision Letter 1]

Dear Dr. Ramasawmy,

We are pleased to inform you that your manuscript 'Genetic analysis of IFNG-AS1 implicates opposite effects to Leishmania guyanensiscutaneous leishmaniasis:  rs4913269 confers protection while rs7134599 enhances susceptibility and correlates with high plasma IL-4 and IL-10 levels' has been provisionally accepted for publication in PLOS Neglected Tropical Diseases.

Best regards,

Vyacheslav Yurchenko, Ph.D.

Academic Editor

Susan Madison-Antenucci

Section Editor

Shaden Kamhawi

co-Editor-in-Chief

Paul Brindley

co-Editor-in-Chief

I thank authors for presenting this interesting work.

---

## [Editor Report · Acceptance letter]

Dear Dr. Ramasawmy,

We are delighted to inform you that your manuscript, "Genetic analysis of IFNG-AS1 implicates opposite effects to Leishmania guyanensiscutaneous leishmaniasis:  rs4913269 confers protection while rs7134599 enhances susceptibility and correlates with high plasma IL-4 and IL-10 levels," has been formally accepted for publication in PLOS Neglected Tropical Diseases.

Best regards,

Shaden Kamhawi

co-Editor-in-Chief

Paul Brindley

co-Editor-in-Chief
